# OpenReview forum: "SafetyAnalyst: Interpretable, transparent, and steerable LLM safety moderation"
_ICLR.cc/2025/Conference — Submitted to ICLR 2025_

### Official Review · Reviewer_Hbfe · 2024-10-29

**Soundness:** 1
**Presentation:** 1
**Contribution:** 1
**Rating:** 1
**Confidence:** 4

**Summary:**

The paper introduces SafetyAnalyst -- although it's not clear why there is also a SafetyReporter as a contribution to the paper -- that produces a harm-benefit-tree and aggregates its features mathematically to accomodate different safety preferences. The authors tackle the current limitations of LLM-based moderation systems in flagging possibly harmful prompts in presence of OOD samples which leads these classification systems astray. The authors claim that their SafetyAnalyst satisfies the two LLM moderation desiderata, i.e., interpretability and steerability, although the experimental section does not necessarily provide further details on the support (or not) of these desiderata.

**Strengths:**

* The scenario looks interesting and challenging and the harm-benefit tree approach where one has LLM teachers such that a smaller student LM (by the way what kind of LM are you using here? BERT-based?) model learns how to "moderate".
* The paper is very suitable for industrial applications rather than being theory-intensive.

**Weaknesses:**

1. It is difficult to interpret the numbers in Table 1. What do they mean? Also SafetyReporter was never mentioned before Table 1. What is the difference between SafetyReporter and SafetyAnalyst?

2. The paper feels as if it was written in haste, where the true insights of the paper are missed for some reason. For example, I would've included a list of contributions right after the introduction section to give the reader a brief summary of what they should expect from the paper.

4. It's unfortunate that all the paper claims made in the method section refer to a table in the appendix.

5. In Section 2.3, the authors claim that they propose a new aggregation algorithm. I have the feeling that this is just a mere multiplication between some predefined weights. Are these weights somewhat learned? Are you also constantly updating $f$, $g$, and $h$ when $W$ and $\gamma$ are updated? This section needs heavy revision.

6. Who's the teacher and who's the student model in Table 2?

7. I makes little sense to say that a model's performance is $F_1 \geq 84.7$. Is the $F_1 = 84.7$ as what it is shown in the table? What are you trying to say in lines 290-292?

8. Although, as per ICLR reviewer's guidelines, not being SoTA for an approach is not reason for rejection, not having at least similar performances isn't great. There's a 6.2 point difference in terms of F1 scores (GPT-4 has 81.6 and SafetyAnalyst has 75.4). This would be justifiable if SafetyAnalyst were more interpretable than the black-box GPT-4. I fail to see why one would prefer SafetyAnalyst rather than GPT-4o.

9. Table 3 reports averages of F1 scores on all datasets. Stating that an average score is better than the other doesn't say much. In fact, if you look at WildGuard and SafetyReporter, the former is constantly better than the latter on each dataset. It just has a performance drop in SORRY-Bench which makes the average F1 plummet. This is a classic scenario where averages aren't trustworthy and, in this scenario, make SafetyAnalyst the second-best performing method after GPT-4, when, instead, WildGuard should be. Here, I would suggest the authors to perform a Friedman test [1] with a post-hoc Bonferroni-Dunn test to assess whether SafetyReporter is better than the rest of the SoTA models without considering GPT-4. Here, one has to use multiple runs over the same dataset -- say 10+ -- to have several F1 scores for each dataset and then perform the Friedman test to tell whether the overall F1 averages are different among the SoTA methods and SafetyReporter. Then, one performs the Bonferroni-Dunn test where the control group is SafetyReporter and assess whether its average F1 score is statistically and significantly different from the rest. For each of the other SoTA methods, except GPT-4, this test gives a p-value which shows us if the method is "better" than the other. Using a p-value of $0.05$ would be sufficient. I expect that SafetyReporter is better than all but WildGuard, which undermines the paper's claim that ''**SafetyReporter outperforms existing LLM safety moderation systems on prompt harmfulness classification.**''

10. Why is the conclusion a brief paragraph? Make it a section where you summarize your paper and future works.

11. The experiments feel cut short where there is no clear connection between the harm-benefit-trees and the performances. Where do harm-benefit-trees come into play here. Do the authors really need these trees? If so, show it somehow. If there is a lack of space, I'd argue that Figure 2 should be rethought. Too much unsupported detail.

12. In the appendix Table 5, it is not clear what the authors are measuring here to assess the agreement with human annotators. Are the authors measuring Cohen's kappa?


[1] Friedman M. The use of ranks to avoid the assumption of normality implicit in the analysis of variance. Journal of the american statistical association. 1937 Dec 1;32(200):675-701.

**Questions:**

See weaknesses.

---

> ### Author Response · Authors · 2024-11-15
> **Clarification of potential misunderstandings**
>
> We sincerely thank Reviewer Hbfe for their detailed feedback and for highlighting areas where our explanations can be improved. However, there appear to have been several misunderstandings, which we address below.
>
> 1. The reviewer asked: _“A smaller student LM (by the way what kind of LM are you using here? BERT-based?)”_. Our base student LM is Llama-3.1-8B-Instruct, which we state multiple times in the main text:
>    - Line 83: “via supervised fine-tuning of Llama-3.1-8B-Instruct”
>    - Line 126: “we fine-tuned an open-weight LM (Llama-3.1-8B-Instruct) to specialize”
>    - Line 207: “LM (Llama-3.1-7B-Instruct) to specialize in the tasks” (typo).
>
> 2. The reviewer stated: _“SafetyReporter was never mentioned before Table 1”_. We introduced SafetyReporter in Figure 1, both in the figure and its captions: _“two specialist models — one to generate harms and one to generate benefits (together named SAFETYREPORTER)”_. Regarding _“What is the difference between SafetyReporter and SafetyAnalyst?”_, please refer to Figure 1’s captions and the paragraph below Table 1 in Section 2.2.
>
> 3. The reviewer claimed: _“All the paper claims made in the method section refer to a table in the appendix”_. We request clarification on _“the method section”_, as our manuscript lacks a section named _“Methods”_. Please specify which of the two appendix tables (Tables 4 and 5) is being referenced.
>
> 4. The reviewer wrote: _“In Section 2.3, the authors claim that they propose a new aggregation algorithm. I have the feeling that this is just a mere multiplication between some predefined weights. Are these weights somewhat learned? Are you also constantly updating $f$, $g$, and $h$ when $W$ and $\gamma$ are updated?”_ Section 2.3 clarifies that the feature weights are fitted to a given label distribution by maximum-likelihood estimation, meaning they are optimized, not predefined or constantly updated. We will revise this section for better clarity.
>
> 5. The reviewer asked: _“Who's the teacher and who's the student model in Table 2?”_ In Line 209, we note _“teacher models (SOTA LLMs)”_, referring to those listed in Lines 76-77: _“SOTA LLMs (GPT-4o, Gemini-1.5-Pro, Llama-3.1-70B-Instruct, Llama-3.1-405BTurbo, and Claude-3.5-Sonnet)”_. Lines 214-215 specify the students as _“The two student models that specialize in harm and benefit feature generation are collectively named ‘SAFETYREPORTER.’”_
>
> 6. The reviewer wrote: _“It makes little sense to say that a model's performance is $F1 \geq 84.7$. Is the $F1 = 84.7$ as what it is shown in the table? What are you trying to say in Lines 290-292?”_ Aggregation models trained on harm-benefit trees generated by all models achieved high classification performance, measured by $F1$, AUPRC, and AUROC, reported in Table 2. We request that the reviewer please clarify why “it makes little sense to say that the model’s performance is $F1≥84.7$”—we used this number since it is the lowest $F1$ among all models reported in Table 2, so the $F1$ scores in Table 2 are all $≥84.7$.
>
> 7. The reviewer asked: _“Why is the conclusion a brief paragraph? Make it a section where you summarize your paper and future works.”_ The Discussion section includes the Conclusion subsection, summarizing our findings and future works. We can rename this section to _“Conclusion”_ if preferred.
>
> 8. The reviewer wrote: _“The experiments feel cut short where there is no clear connection between the harm-benefit-trees and the performances. Where do harm-benefit-trees come into play here. Do the authors really need these trees?”_ Sections 2.3 and 2.4 detail how the features in the harm-benefit tree are aggregated numerically and translated into a safety label that was used for evaluation in our experiments (i.e., the harm-benefit tree serve as the input to the aggregation algorithm, which outputs a harmfulness score that is then converted into a binary label). Nonetheless, we agree that further studies showing the usefulness of different features in the harm-benefit trees would be compelling, so we are working on supplementing the manuscript with ablations of different types of features in the trees (e.g., actions, effects, etc.).
>
> 9. The reviewer wrote: _“In the appendix Table 5, it is not clear what the authors are measuring here to assess the agreement with human annotators.”_ Lines 814-816 explain: _“To obtain the agreement rates, we computed the proportion of positive ratings (e.g., very plausible, somewhat plausible, and reasonable) among all positive and negative ratings.”_ We will move this explanation to the table captions for clarity.
>
> We hope this clarification resolves the misunderstandings and respectfully ask Reviewer Hbfe to reconsider their assessment in light of this explanation while we work on revising the manuscript. Once we have updated the manuscript, we will comment again with a point-by-point response to all reviewers’ comments. Meanwhile, we appreciate your time and effort in revisiting the above points.

---

> ### Comment · Reviewer_Hbfe · 2024-11-19
> **Cool. Some more stuff.**
>
> 1. Cool.
> 2. In my experience, you shouldn't introduce methods in captions. There isn't any single place where SafetyReporter is in the text before the Figure. Please introduce SafetyReporter in the text as well. What is the difference between SafetyReporter and SafetyAnalyst? Please respond here concisively.
> 3. It is still a methods section, although the title of it is not "Method" per se. I'm referring to both Tables. However, we're not here to discern useless stuff (i.e., "hey which Table is the reviewer referring to"). My concern is that all the claims of novelty are deferred to the appendix, which makes the paper wierd to read.
> 4. Cool.
> 5. Cool.
> 6. I'd still go with an F1 equals to bla bla bla here. It confuses the reader to see a $\geq$. But, thanks for the clarification.
> 7. Yeah, rename it, and extend it to at least 2 paragraphs.
> 8. Cool, I'd like to see these ablations before the discussion period ends. Is this feasible?
> 9. Nice. Repeating stuff in the table caption is very useful when parsing the table in isolation to the text.

---

> ### Comment · Reviewer_Hbfe · 2024-11-19
> **What about my previous suggestions?**
>
> 1. Table 3 reports averages of F1 scores on all datasets. Stating that an average score is better than the other doesn't say much. In fact, if you look at WildGuard and SafetyReporter, the former is constantly better than the latter on each dataset. It just has a performance drop in SORRY-Bench which makes the average F1 plummet. This is a classic scenario where averages aren't trustworthy and, in this scenario, make SafetyAnalyst the second-best performing method after GPT-4, when, instead, WildGuard should be. Here, I would suggest the authors to perform a Friedman test [1] with a post-hoc Bonferroni-Dunn test to assess whether SafetyReporter is better than the rest of the SoTA models without considering GPT-4. Here, one has to use multiple runs over the same dataset -- say 10+ -- to have several F1 scores for each dataset and then perform the Friedman test to tell whether the overall F1 averages are different among the SoTA methods and SafetyReporter. Then, one performs the Bonferroni-Dunn test where the control group is SafetyReporter and assess whether its average F1 score is statistically and significantly different from the rest. For each of the other SoTA methods, except GPT-4, this test gives a p-value which shows us if the method is "better" than the other. Using a p-value of would be sufficient. I expect that SafetyReporter is better than all but WildGuard, which undermines the paper's claim that **SafetyReporter outperforms existing LLM safety moderation systems on prompt harmfulness classification.**
>
>
> 2. There's a 6.2 point difference in terms of F1 scores (GPT-4 has 81.6 and SafetyAnalyst has 75.4). This would be justifiable if SafetyAnalyst were more interpretable than the black-box GPT-4. I fail to see why one would prefer SafetyAnalyst rather than GPT-4o.

---

> > ### Author Response · Authors · 2024-11-27
> > **Revised manuscript and point-by-point response to the reviewer's comments (Part 1)**
> >
> > We thank Reviewer Hbfe for their prompt response and valuable feedback. We have conducted additional analyses and revised the manuscript to address the reviewer’s constructive comments. All changes are highlighted in red in the revised manuscript. We hope these updates strengthen our work and respectfully encourage the reviewer to consider raising their rating. Below are detailed responses to the reviewer’s comments:
> >
> > ## 1. SafetyAnalyst vs. SafetyReporter
> >
> > We apologize for any confusion regarding these terms. **SafetyAnalyst** is an abstract *framework* using chain-of-thought prompting to generate harm-benefit trees that can be aggregated into safety labels using a mathematical algorithm. **SafetyReporter** is the specific *system* we implemented (including a pair of knowledge-distilled specialist LMs that produce harm trees and benefit trees, as well as a particular mathematical algorithm optimized on a particular dataset) using the framework for prompt safety classification. The distinct names highlight the framework’s general applicability, allowing for alternative implementations and aggregation methods. This distinction is clarified in the **Abstract**, **Contributions** paragraph at the end of **Section 1**, and in **Figure 1**.
> >
> > ---
> >
> > ## 2. Main Claims and Supplementary Tables
> >
> > Our main claims are not dependent on the tables in the appendices. They are:
> > 1. The pipeline of the **SafetyAnalyst** framework (**Fig 1**).
> > 2. The extensiveness of generated safety features (**Fig 2** and **Table 4**, which is now in Appendix B for space).
> > 3. The interpretable, transparent, and steerable decision-making mechanism (**Sections 2.3, 2.4; Fig 3**).
> > 4. The competitive performance of **SafetyReporter** on prompt safety classification (**Tables 1, 2**).
> >
> > The other appendix tables provide auxiliary information, such as details on the number of harm-benefit trees we collected per teacher model and prompt dataset (**Table 3**) and human agreement rates on different types of features in the harm-benefit trees generated by different LMs (**Table 5**).  We hope that the **Contributions** paragraph at the end of **Section 1** helps to clarify this.
> >
> >
> > ---
> >
> > ## 3. Statistical Testing of SafetyReporter vs. WildGuard
> >
> > We appreciate the suggestion to include additional statistical tests comparing **SafetyReporter** and **WildGuard**. However, we are less sure that the specific test recommended by the reviewer is the best way to go:
> >
> > 1. **Deterministic Outputs**
> >    In line with standard LLM safety evaluation practices, all baselines were evaluated with a sampling temperature of 0, producing deterministic outputs. Consequently, re-running the evaluation on the same dataset yields identical F1 scores, invalidating statistical procedures that assume variance between repeated measurements.
> >
> > 2. **Unequal Benchmark Sizes**
> >    The reviewer suggested treating each benchmark as a measurement for statistical testing. However, this assumption conflicts with the highly uneven sizes of benchmarks (e.g., 100 prompts in SimpleSafetyTests vs. 9,450 in SORRY-Bench). Testing at the benchmark level would unfairly overweight smaller benchmarks.
> >
> > However, we did want to address the reviewer’s concern—we performed a chi-square test on classification accuracy across all prompts, treating prompts from different benchmarks equally. The chi-square statistic is 41.0186 with p $<$ 0.00001, suggesting that **SafetyReporter** significantly outperformed **WildGuard** overall.

---

> > > ### Author Response · Authors · 2024-11-27
> > > **Revised manuscript and point-by-point response to the reviewer's comments (Part 2)**
> > >
> > > ## 4. Advantages of SafetyReporter over GPT-4
> > >
> > > We clarified **SafetyReporter**’s advantages over **GPT-4** despite a lower F1 score:
> > >
> > > 1. **Enhanced Interpretability and Steerability**
> > >    We added a new section —**Section 3.3** — to emphasize **SafetyReporter**’s advantages, highlighting:
> > >    > This interpretability is two-folded: first, the features, on which the safety decisions are based solely, are explicitly generated by SAFETYREPORTER and semi-structured (i.e., on carefully curated dimensions, including stakeholder, harm, benefit, action, effect, extent, likelihood, and immediacy); second, these features are aggregated using a white-box algorithm with transparent mechanisms and interpretable feature weights that quantify the importance of corresponding feature values (Figure 3). Even though LLMs (such as GPT-4) can generate explanations for their decisions, there remains a lack of interpretability in how the decisions are reached and there is no reliable causal relationship between the explanation and the safety prediction.
> > >
> > >    > SAFETYREPORTER’s aggregation algorithm is defined by a set of transparent, interpretable parameter weights. The weights of the parameters we report in Figure 3 reflect the values of the annotators who provided the labels for the WildJailbreak dataset, for which the algorithm was optimized. However, one central strength of the SAFETYANALYST approach is that the aggregation algorithm allows different safety features to be up- or down-weighted for top-down adjustments, or fitted to a customized safety label distribution for bottom-up adjustments (e.g., personalized safety alignment). Bottom-up adjustments of weights can be achieved by fitting the aggregation model to a safety label distribution produced by an individual or group; the resulting parameters would be aligned to the values expressed in the labels. We provide concrete explanations for how to operationalize top-down weight adjustments in the case study in Appendix E.
> > >
> > > 2. **Case Study in Appendix E**
> > >    We added a case study demonstrating **SafetyReporter**’s transparency, interpretability, and steerability. The example illustrates how prompts are processed and how feature weights can be adjusted to align with different safety standards.
> > >
> > > 3. **Open-Source Advantages**
> > >    **SafetyReporter** is an open-source, lightweight, and cost-effective system, fostering open AI research, unlike proprietary models including GPT-4.
> > >
> > > ---
> > >
> > > ## 5. Additional Changes
> > >
> > > We incorporated changes in the manuscript:
> > > - Added a **Contributions** paragraph (end of **Section 1**)
> > > - Revised **Sections 2.3 and 2.4** to improve clarity on the aggregation model
> > > - Avoided using $\geq$ before F1 scores
> > > - Renamed **Discussion** to **Conclusion**
> > > - Provided ablation studies in **Appendix D.4** to provide additional context for inference time (**Section 2.3**)

---

### Official Review · Reviewer_hQ2Z · 2024-11-02

**Soundness:** 2
**Presentation:** 2
**Contribution:** 1
**Rating:** 3
**Confidence:** 4

**Summary:**

The SAFETYANALYST framework is a system for moderating content that aims to be both interpretable and adaptable to specific values. It uses harm-benefit trees to evaluate actions and their effects, producing a harmfulness score that aligns with safety preferences, and it outperforms existing systems on prompt safety benchmarks. They have considered various stakeholders and compared the results with five well-known LLM solutions.

**Strengths:**

The use of both harm-benefit trees and symbolic knowledge distillation is the key element in this research.

**Weaknesses:**

The idea is promising and the paper is focusing on a major issue of AI Safety that is scientifically sounds. Still, there is room to improve it to reach a high level of originality, quality, clarity, and significance. The following comments can be addressed to improve the paper from all mentioned aspects:
1- The literature review of the paper could be improved, there are various papers on LLM safety. For example, "SafeLLM", "TrustLLM", etc and most of them focus on the same issue. For example, SafeLLM goes even deeper and focuses on Domain-specific or in-context safety.
2- The author must provide results regarding computation complexity and delay in response time in the provided model.

**Questions:**

1- How the proposed solution is robust again jail breaking and prompt injection.
2- Considering the problem of in-context reward hacking, how the proposed method could help us to avoid such issues?
3- I think more in depth research is needed to gain a proper novelty and originality. For example, one may consider formation of concepts in LLMs and try to fix the issue in the that level considering research like this: https://www.anthropic.com/research/mapping-mind-language-model

---

> ### Author Response · Authors · 2024-11-27
> **Revised manuscript and point-by-point response to the reviewer's comments**
>
> We sincerely thank Reviewer hQ2Z for their thoughtful feedback and detailed critique of our manuscript. We have carefully considered all comments and conducted additional analyses, revising the manuscript accordingly. All changes are highlighted in red in the revised manuscript. We hope these updates address the reviewer's concerns and strengthen the manuscript. Below, we provide a point-by-point response to the reviewer's comments:
>
> ---
>
> 1. **Literature Review**
>    The reviewer recommended improving the literature review. To address this, we have expanded **Section 4** to include references to additional relevant work, such as *TrustLLM* and *SafeLLM*.  In addition, given that our system is designed to be an innovation in safety content-moderation, we now add detailed descriptions of the other comparable safety content-moderation systems in **Appendix D.1**. These additions provide a broader context and further situate our contributions within the existing body of research.
>
> ---
>
> 2. **Inference Time Evaluation**
>    We have reported inference time evaluation results for **SafetyReporter** and **WildGuard** in **Section 3.2** of the revised manuscript. To address concerns about improving inference time for **SafetyReporter**, we recommend reducing the complexity of the harm-benefit tree. To support this, we provide ablation studies (**Appendix D.4**) on the features of the harm-benefit tree in **WildJailbreak** and **WildGuardTest**, showing how these features contribute to feature aggregation. We acknowledge that these results may not generalize across all datasets or use cases, and, thus, include all features in the harm-benefit tree in the current manuscript for generality.
>
> ---
>
> 3. **Robustness Against Jailbreaking and Prompt Injection**
>    The reviewer inquired about the robustness of the proposed solution against jailbreaking and prompt injection attacks. To address this, we highlight that **SafetyReporter** was fine-tuned on harm-benefit trees generated from both vanilla and *adversarial* prompts sampled from *WildJailbreak*. This dataset includes synthetic vanilla harmful and benign prompts as well as adversarial prompts created using various combinations of jailbreaking techniques. The performance of **SafetyReporter** on the adversarial subset of *WildGuardTest* demonstrates its effectiveness against adversarial attacks (see **Table 2**; 4th column from the left). Furthermore, in **Appendix D.3**, we provide a breakdown of classification accuracy for **SafetyReporter** and other baselines on fine-grained prompt categories in *SORRY-Bench*. Here, **SafetyReporter** exhibited the highest robustness to persuasion techniques including Authority Endorsement, Evidence-based Persuasion, Expert Endorsement, Logical Appeal, and Misrepresentation.
>
> ---
>
> 4. **In-Context Reward Hacking**
>    The reviewer asked, “Considering the problem of in-context reward hacking, how the proposed method could help us to avoid such issues?” We kindly request clarification on what is meant by “in-context reward hacking” in this context. To our understanding, the **SafetyReporter** framework does not incorporate a reward model or RLHF, so it is unclear how this issue directly applies to our work. We are happy to address this point further once additional clarification is provided.
>
> ---
>
> 5. **Clarification on Novelty**
>    The reviewer suggested leveraging mechanistic interpretability to increase the novelty of our work — an example of which is Anthropic’s research into surfacing concepts latent in their language model. We agree that using mechanistic interpretability to form “concepts” and enhance LMs’ reasoning is an interesting and important way of increasing the transparency and interpretability of AI systems. However, our approach leverages a different and completely novel (and computationally cheaper) method of making an LM-based content moderation system more interpretable.  In our approach, we employ chain-of-thought prompting over a semi-structured harm-benefit feature space to generate harm-benefit trees associated with extents, likelihoods, and severities, which are then aggregated by an interpretable algorithm.  All of this guides the model's reasoning about prompt safety, as illustrated in **Figures 1 and 2**. To emphasize the novelty and contributions of our work, we have added a **Contributions** paragraph to the end of **Section 1** and expanded on the interpretability, transparency, and steerability of our system compared to other baselines in **Section 3.3**.
>
> ---
>
> We hope these updates address the reviewer’s concerns. We are grateful for the constructive feedback and respectfully encourage the reviewer to consider raising their rating.

---

### Official Review · Reviewer_SruY · 2024-11-04

**Soundness:** 3
**Presentation:** 3
**Contribution:** 3
**Rating:** 6
**Confidence:** 2

**Summary:**

The authors introduce SafetyAnalyst, a language model solution to LLM content moderation. Critically, SafetyAnalyst is both interpretable and tunable to reflect different safety values while achieving SOTA performance on prompt safety benchmarks. The authors achieve this by training two open-weight LM to identify trees of stakeholders, actions, and effects from text prompts. One focuses on harms and the other on benefits. The model output is tunable to different safety values via a parameterized feature aggregation algorithm for calculating harmfulness. The output is interpretable due to the generated harm-benefit trees which consist of chains of stakeholders to actions to effects.

**Strengths:**

The authors introduce a novel interpretable moderation system which achieves SOTA performance in a dense field. SafetyReporter provides structured output which helps humans understand the harmfulness of a prompt.
The paper presents a wealth of both benchmark datasets and models. This provides clear evidence that SafetyReporter is strong in a wide variety of content moderation.
The SafetyAnalyst framework is clearly explained. The authors make great use of graphics explaining the overall framework and the role of the language model. Additionally, the discussion of value alignment displays the flexibility of the system.

**Weaknesses:**

In the related work section there is room for more discussion on the differences between existing content moderation systems. Both how the baseline models differ from each other and how they differ from SafetyReporter and SafetyAnalyst as a whole. Discussion beyond “[existing content moderation systems] internal decision mechanisms are challenging to interpret” would strengthen the authors’ claims about the importance of interpretability in content moderation.
More experiments on the steerability of the content moderation system would be beneficial. As the paper stands, the authors do a good job explaining how to align the system with a dataset. However, experimentation about how alignment would make the system more effective for specific tasks is missing. For example, the authors could align the aggregation weights of SafetyAnalyst to a held-out portion of a benchmark dataset and look at how performance improves.

**Questions:**

Overall the paper is strong however a few areas could use clarification.
In the limitation section the authors discuss the tradeoff between interpretability and inference time. Quantifying the difference in inference time between baselines, SafetyAnalyst, and LLMs would be beneficial to weigh the value of this tradeoff.
Additionally, more insight on the implications of GPT4 outperforming both SafetyAnalyst and baselines would be helpful.
In the evaluation results section the authors’ mention that SafetyReporter was not aligned to any of the benchmark datasets. How does performance differ if alignment is done? Additionally, I would be interested in how aggregation feature weights change from baseline to baseline.

---

> ### Author Response · Authors · 2024-11-27
> **Revised manuscript and point-by-point response to the reviewer's comments**
>
> We sincerely thank Reviewer SruY for their positive feedback and thoughtful critique of our manuscript. We have conducted additional analyses and revised the manuscript accordingly to address the reviewer's insightful comments. All changes are highlighted in red in the revised manuscript. We hope these updates strengthen our work and respectfully encourage the reviewer to consider raising their rating. Below, we provide detailed responses to each of the reviewer’s comments:
>
> ---
>
> ### 1. Request for more discussion on the differences between existing content moderation systems
>
> The reviewer suggested providing further discussion on the differences between existing content moderation systems. To address this, we have expanded the subsection titled “Existing LLM content moderation systems” in Section 4 and added **Section 3.3** in the main text, which elaborates on the advantages of interpretability, transparency, and steerability in our system compared to the baselines. Additionally, we included detailed descriptions of the baseline models in **Appendix D.1**, where we highlight their differences and contextualize them in relation to our system.
>
> ---
>
> ### 2. Evaluation of steerability and alignment to different datasets
>
> We agree with the reviewer that providing a demonstration of how our system can be aligned would bolster our claims of steerability. In response to this concern, we now demonstrate the capacity of the model to be steered using a case study (see **Appendix E**).
>
> In future work, we are eager to demonstrate steerability more comprehensively.  However, we encountered a few blockers when attempting to tackle this challenge using the safety moderation benchmarks that are included in the paper (and others like them).  Existing prompt safety datasets are designed to cover many safety categories comprehensively and capture safety intuitions that the majority of people share.  That is, both the prompts and the responses are drawn from similar distributions, so the benefit of steering a content moderation system to align to one of these datasets over another is rather limited.  On the other hand, different individuals are likely to make at least somewhat different judgments across a range of safety-related cases and we do expect that our system should be able to be steered to capture individual-level variation.  However, extant prompt safety datasets lack identifying information that would allow us to determine which answers are given by which participants.  Given these limitations, adequately demonstrating the steerability of SafetyReporter fully would require substantial new annotation data.  This is an exciting direction for future work.
>
> ---
>
> ### 3. Inference time evaluation results
>
> We have reported inference time evaluation results for **SafetyReporter** and **WildGuard** in **Section 3.2** of the revised manuscript. To address concerns about improving inference time for **SafetyReporter**, we recommend reducing the complexity of the harm-benefit tree. To support this, we provide ablation studies (**Appendix D.4**) on the features of the harm-benefit tree in **WildJailbreak** and **WildGuardTest**, showing how these features contribute to feature aggregation. We acknowledge that these results may not generalize across all datasets or use cases, and, thus, include all features in the harm-benefit tree in the current manuscript for generality.
>
> ---
>
> ### 4. Performance insights on SORRY-Bench
>
> The reviewer inquired about the reasons behind performance differences between **GPT-4** and **SafetyReporter**, which was driven by **SORRY-Bench**. In **Appendix D.3**, we offer a detailed breakdown of classification accuracy across different models and prompt categories:
>
> - **GPT-4** outperformed **SafetyReporter** by successfully detecting a subset of Encoding and Encrypting prompts (Atbash and Caesar ciphers).
> - **SafetyReporter** demonstrated the most robustness to persuasion techniques (Authority Endorsement, Evidence-based Persuasion, Expert Endorsement, Logical Appeal, and Misrepresentation).
> - **WildGuard** failed to identify potentially unsafe prompts in certain non-English categories (Marathi, Malayalam, and Tamil).
>
> These observations are summarized in **Appendix D.3** to provide clarity on the comparative strengths and weaknesses of the models.  Note that it would be possible to prompt or train **SafetyReporter** to look for and decode ciphers if that were of interest for a particular use case — and which would improve performance on SORRY-Bench.
>
> ---
>
> We hope that these revisions and clarifications adequately address the reviewer’s comments. Thank you again for your valuable feedback, which has greatly helped us improve the quality of our work.

---

### Meta-Review · Area_Chair_Tywc · 2024-12-16

**Metareview:**

After reading the reviewers' comments, and reviewing the paper, I regret to recommend rejection.

The paper introduces SafetyAnalyst, an interpretable and tunable language model solution to LLM content moderation.
Two critical issues of the contribution stems from lack of proper literature review and assessment of competing work as well as the relatively poor presentation of the paper, that is not easy to follow in several places.

The authors may consider the following points to improve their contribution:

1.	Expand the competing work and comparison with existing literature.
2.	Significantly improve readability.
3.	Clearly explain novelty compared to existing literature.

**Additional Comments On Reviewer Discussion:**

The authors were proactive in responding to the reviewers' comments; reviewer Hbfe was engaged in responding to the authors.

No ethics review raised by the reviewers, and we agree with them.

---

### Decision · Program_Chairs · 2025-01-22

Reject